# A Metabolomics and Big Data Approach to Cannabis Authenticity (Authentomics)

**DOI:** 10.3390/ijms24098202

**Published:** 2023-05-03

**Authors:** Pramodkumar D. Jadhav, Youn Young Shim, Ock Jin Paek, Jung-Tae Jeon, Hyun-Je Park, Ilbum Park, Eui-Seong Park, Young Jun Kim, Martin J. T. Reaney

**Affiliations:** 1Department of Food and Bioproduct Sciences, University of Saskatchewan, Saskatoon, SK S7N 5A8, Canada; pramodkumar.jadhav@usask.ca; 2Prairie Tide Diversified Inc., Saskatoon, SK S7J 0R1, Canada; 3Department of Food and Biotechnology, Korea University, Sejong 30019, Republic of Korea; yk46@korea.ac.kr; 4Herbal Medicines Research Division, Ministry of Food and Drug Safety, Cheongju 28159, Republic of Korea; 5Yuhan Care R&D Center, Yuhan Care Co., Ltd., Yongin 17084, Republic of Korea; 6Yuhan Natural Product R&D Center, Yuhan Care Co., Ltd., Andong 36618, Republic of Korea

**Keywords:** cannabis authenticity, *Cannabis sativa* L. authentomics, metabolites, NMR

## Abstract

With the increasing accessibility of cannabis (*Cannabis sativa* L., also known as marijuana and hemp), its products are being developed as extracts for both recreational and therapeutic use. This has led to increased scrutiny by regulatory bodies, who aim to understand and regulate the complex chemistry of these products to ensure their safety and efficacy. Regulators use targeted analyses to track the concentration of key bioactive metabolites and potentially harmful contaminants, such as metals and other impurities. However, the metabolic complexity of cannabis metabolic pathways requires a more comprehensive approach. A non-targeted metabolomic analysis of cannabis products is necessary to generate data that can be used to determine their authenticity and efficacy. An authentomics approach, which involves combining the non-targeted analysis of new samples with big data comparisons to authenticated historic datasets, provides a robust method for verifying the quality of cannabis products. To meet International Organization for Standardization (ISO) standards, it is necessary to implement the authentomics platform technology and build an integrated database of cannabis analytical results. This study is the first to review the topic of the authentomics of cannabis and its potential to meet ISO standards.

## 1. Introduction

Metabolomics is a crucial approach for gaining insight into the largest possible set of low-molecular-weight metabolites present in biological samples. When used in conjunction with genomics, transcriptomics, and proteomics, metabolomics helps shed light on the workings of biological systems as they develop and respond to environmental stimuli.

Metabolomics is downstream of genomics, transcriptomics, and proteomics [1,2]. Comprehensive analysis of the metabolome is predicated on developments in analytical methods, data-handling tools, and database management systems that first generate big data sets using various chemometric techniques and subsequently use multivariate analysis for interpretation [3]. Nuclear magnetic resonance (NMR) and chromatography (gas or liquid) coupled with mass spectrometry (MS) are the most common techniques used in metabolomic analysis. There are different approaches in metabolomics for the comprehensive analysis of both known and unknown metabolites. One approach, metabolic profiling, involves measuring large sets of metabolites to provide information about metabolism. Such an approach can include the characterization of both metabolites (unknown and known) and metabolic pathways. Analytical methods that focus on the repeated identification and quantification of pre-selected compounds are known as targeted approaches. On the other hand, non-targeted approaches quantify all measurable compounds, regardless of their identification. Both targeted and non-targeted methods can provide information about the concentration of known compounds, while unknown compounds can be interpreted as having relative concentrations. A third approach, called metabolic fingerprinting, typically generates metabolic information without precise quantification and identification. This latter approach involves the production of a pattern that is, ideally, interpretable. Fingerprinting is used in food or food product authentication, where the fingerprint pattern of the unknown sample is compared with the spectral database of known samples to determine its conformity [4,5]. Metabolic studies can also be classified based on the study objectives, such as (a) informative studies, where the metabolites’ identification and quantification are obtained; (b) discriminative studies, which help to distinguish metabolites among sample populations; and (c) predictive studies, which create statistical models to create class memberships [6].

Cannabis and its extracts are chemically complex natural mixtures with various biologically active compounds (metabolites). These compounds include phytocannabinoids, terpenoids, flavonoids, nitrogenous compounds, sugars, proteins, fatty acids, and more (Table 1).

There were 423 compounds reported in the 1980s [7], and in 2017, the number of identified compounds increased to ~565, and more are being identified. The number of cannabinoids detected is now over 120 [8,9,10,11]. Among these cannabinoids, psychoactive Δ^9^-tetrahydrocannabinol (THC) and cannabidiol (CBD) are best known for their contribution to pharmacological activity (Figure 1). These compounds are not the products of metabolic pathways but rather are produced as acidic precursors. The action of heat on these precursors induces decarboxylation and the formation of bioactive compounds. Cannabis extracts contain a range of compounds, including cannabinoids and terpenes, of which many contribute a synergistic “entourage effect” where the therapeutic effect is greater than the sum of the individual compounds [12]. While the US Food and Drug Administration (FDA)-approved drugs such as Epidiolex, Marinol, Syndros, and Cesamet have demonstrated efficacy in treating certain ailments, there is still a great deal of potential for the use of cannabinoids in treating other conditions. Current research suggests that cannabinoids may have anti-inflammatory, analgesic, and anxiolytic properties, among others. However, further clinical trials are needed to fully understand the therapeutic potential of these compounds and mixtures of compounds. Interestingly, cannabinoids bind to receptors and exert biological effects [13]. As one might expect, one class of receptors—cannabinoid receptors—was identified through their interaction with cannabinoids.

Metabolomics analysis is useful in studies of plant responses to their environment (e.g., temperature, photoperiod, bioelicitors, fertilizers, water, atmosphere, etc.) and genotypic differences among plants. For example, metabolomics approaches can be used in developing better agronomic practices or the selection of cultivars with superior traits. Gas chromatography (GC) with a flame ionization detector (FID) or mass spectrometer is often used in the analysis of cannabis and cannabis extracts. However, these approaches are limited to measuring metabolites that can be made volatile. Acidic precursors of THC and CBD experience decarboxylation in the typical GC injection port. If this is not carefully controlled, the analysis will be compromised.

Acidic cannabinoids can be stabilized and made more volatile by derivatization, especially via silylation, but the quantification can be less reliable [14]. Moreover, in the high-temperature conditions typical in a GC injector, cannabinoids can thermally oxidize or isomerize. For example, unnatural compounds produced by isomerization, Δ^8^-THC and cannabinol (CBN), were detected in cannabinoid extract analysis by GC. High-performance liquid chromatography (HPLC) analysis is possible for the non-destructive analysis of cannabinoids and compounds can be resolved using a range of media, though reverse-phase columns are widely used. While GC methods typically provide better resolution than HPLC, peak overlap in HPLC can typically be overcome when an MS detector is used. When MS/MS is applied, fragmentation patterns can serve to definitively identify cannabinoids [14]. NMR has been used in cannabis extract analysis to discriminate among cultivars and determine the impact of elicitors in cannabis cell suspension cultures. NMR has advantages over chromatographic methods, including simplified sample preparation and non-destructive analysis. These characteristics make some NMR-based analyses suitable for high-throughput analysis and more reproducible than other methods. NMR is much less sensitive than MS but can provide, with fewer sample preparation steps, robust information regarding chemical fingerprints [15,16]. NMR methods also have an additional capability that is not always possible with other methods. NMR methods can be linear over a much larger dynamic range and interference can often be managed. In MS methods, the response is based on ionization phenomena, and the ion suppression of signals is common. Quantitative MS is difficult to accomplish without sophisticated standards for each analyte.

## 2. Metabolomic Technologies

Many analytical tools have been applied to extract useful metabolomic information. However, due to the chemical heterogeneity of metabolites, large differences in metabolite concentration, and interactions among metabolites, single analytical platforms may fail in determining metabolic profiles. Therefore, combinations of analytical approaches are needed to capture most of the salient information required to characterize complex mixtures of compounds. The selection of the best analytical solution is influenced by the sample matrix, metabolite concentration and properties, and sample amount. Thus, metabolomics is described as an area of science rather than an analytical approach [17]. The metabolomic technologies have been used to identify bioactive compounds in cannabis and are summarized in Table 2 and briefly described below.

### 2.1. NMR

NMR has been used as a metabolic fingerprinting tool to identify and characterize metabolites in plant extracts. Choi et al. (2004) [16] performed the metabolomic analysis of 12 *C. sativa* cultivars using proton NMR (^1^H NMR) and analyzed the data using multivariate analysis techniques. The ground cannabis material was extracted with 50% methanol and chloroform and both water-soluble and chloroform-extracted fractions were collected and separated for further analysis. The water extract was enriched in primary metabolites including carbohydrates (glucose and sucrose) and amino acids (asparagine and glutamic acid). ^1^H-^1^H correlation spectroscopy (COSY) and total COSY (TOCSY) spectra were used to assign the residual proton signals of sugars, and Heteronuclear multiple bond correlation (HMBC) spectra gave evidence regarding amino acids present. Higher levels of carbohydrates and lower amino acid content were detected in leaves than in flowers. However, non-polar metabolites such as cannabinoids (THC, ∆^9^-tetrahydrocannabinolic acid (∆^9^-THCA), CBD, cannabidiolic acid (CBDA), and CBN) were detected primarily in chloroform-extracted fractions. Principal component analysis (PCA, covariance method) was used to discriminate among cultivars using THCA and CBDA as major metabolites. In addition, water extracts containing amino acids and carbohydrates can also be used for identification. This technique uses limited information regarding metabolites to distinguish cultivars [16].

**Table 2 ijms-24-08202-t002:** Different technologies for the identification of compounds in cannabis.

Identification	Matrix	Cultivar/Strain	Conditions	Products	Refs
*NMR*	*A metabolic fingerprinting tool to identify and characterize metabolites in plant extracts*				
^1^H NMR	Flowers, leaves	*Cannabis sativa* L.	Recorded on 400.13 MHz in CDCl_3_ and D_2_O, plant material was extracted with 50% aqueous methanol and chloroform.	THC, THCA, CBD, CBDA, CBN	[16]
^1^H NMR + RT-PCR	Trichomes, Flowers, leaves	Bedrocan, Bedica	Recorded on 500.13 MHz. Fresh materials were ground to a fine powder using a pestle and a mortar under cold conditions.	THCA	[18]
^1^H NMR	Flowers, leaves, stalks	*Cannabis sativa* L.	Recorded at 300 K and 400 MHz and performed in DMSO-D_6_ without internal standards.	Total THC or the sum of THC, THCA, and CBN	[19]
*GC-FID/MS*	*Metabolomic approaches to quantify and identify cannabinoids and terpenes*				
GC-FID/MS	Flowers	*Cannabis sativa* L., Indica, hybrid, Bedrocan	Extracted with absolute ethanol and 1-octanol.	8 Major neutral cannabinoids and 36 terpenes	[20]
GC-FID/MS	Flowers	Bedrocan, Bedropuur, Bediol	Peak area variation of the internal standard 1-octanol for all cannabis samples.	9 cannabinoids and 27 terpenoids	[21]
GC-MS	Seeds	*Cannabis sativa* L.	The crushed seed was extracted with methanol and centrifuged with ribitol as an internal standard.	236 Untargeted metabolites were identified	[22]
GC × GC	Flowers	*Cannabis sativa* L., Indica, hybrid	The extraction of cannabis flower samples with a solvent mixture (water/methanol/acetone) with a stir bar coated with polydimethylsiloxane.	Monoterpenes, sesquiterpenes, hydrocarbons, cannabinoids, terpenoid alcohols, and fatty acids	[23]
GC-MS/MS	Flowers	Medical cannabis strain	Ground samples were subjected to direct measurement with a static headspace. sampler, using a semi-polar stationary phase GC column.	93 Terpenoids	[24]
GC-MS	Flowers	*Cannabis sativa* L.	Used for the profiling of cannabis because of its sensitivity and ability to highlight the aromatic expression of chemovars.	67 Terpenes (29 monoterpenes and 38 sesquiterpenes)	[25]
*LC-MS*	*Applicable to cannabis complex mixtures, polar and non-polar compounds*				
ESI-LC/MS	Flowers	Medical cannabis strain	Extracted with ethanol, producing fractions.	94 Cannabinoids	[26]
UHPLC	Flowers	*Cannabis sativa* L.	Extracted with 80% ethanol.	CBD and THC	[27]
LC-MS/MS, LC-QTOF	Plant	*Cannabis sativa* L.	Extracted using supercritical CO_2_ with ethanol as a cosolvent. Atmospheric pressure chemical ionization with multiple reaction monitoring was used to quantify cannabinoids.	6 Major cannabinoids and seven minor cannabinoids, CBD, CBN, and THC	[28]
LC-HRMS/MS	Flowers	Bediol	Analyzed using a non-targeted metabolomics approach. Isocratic elution with water/MeCN 30:70 and 0.1% formic acid.	CBD, CBDA, THC, THCA, CBGA, and CBN	[29]
GC-TOF/MS, LC-QTOF MS/MS	Flowers, leaves	*Cannabis sativa* L.	Polar and non-polar cannabis extracts.	134 in the non-polar extracts and 46 in the polar extracts	[30]
LC-HRMS, UV-C treatment	Leaves	*Cannabis sativa* L.	Extracted with isopropanol, filtered through SPE C_18_ columns, and subjected to LC-TOF/MS analysis.	Not for cannabinoid content	[31]
UPLC/ESI (+) and (–) modes	Flowers	Medical cannabis strain	Dried cannabis was extracted with supercritical fluid at ambient temperature to obtain a native extract. The native extract was subjected to heat to prepare the decarboxylated extract.	62 compounds including 23 phytocannabinoids, terpenoids, flavonoids, hydrocarbons, phenols, and fatty acids	[32]
*Others*	*Employed for spectral fingerprinting of cannabis samples*				
TD-IMS	Flowers, leaves	*Cannabis sativa* L.	Powdered cannabis material was extracted with hexane and centrifuged.	CBD, CBDV, CBG, THC, THCV and acidic forms	[33]
TLC	Flowers	Bedrocan, Bediol	Developed and validated with the use of pure cannabinoid reference standards and two medicinal cannabis cultivars.	CBD, THC, THCV, CBG, and CBC	[34]
HCARS	Flowers	Bedrobinol, Fedora	Identified and localized THCA or CBDA and myrcene in secretory cavities of drug-type and fiber-type glandular trichomes.	THCA or CBDA and myrcene	[35]
STELDI-MS	Leaves	*Cannabis sativa* L.	Extracted using water–methanol solvents then applied by spotting onto a silica gel 60 plate.	CBD, CBN, THC, CBC, CBDB, CBCVA, and CBDVA	[36]

Abbreviations: ^1^H NMR, proton nuclear magnetic resonance; RT-PCR, real-time polymerase chain reaction; CBDA, cannabidiolic acid; CBGA, cannabigerolic acid; CBCVA, cannabichromevarinic acid; CBDVA, cannabidivarinic acid; CBC, cannabichromene; CBD, cannabidiol; CBG, cannabigerol; CBN, cannabinol; THC, tetrahydrocannabinol; THCA, tetrahydrocannabinolic acid; THCV, tetrahydrocannabivarin; GC-MS, gas chromatography coupled with mass spectrometry detector; FID, flame ionization detector; HPLC-MS, high-performance liquid chromatography coupled with mass spectrometry; ESI, electrospray ionization; UPLC, ultra-performance liquid chromatograph; TD-IMS, thermal desorption ion mobility spectrometry; TLC, thin-layer chromatography; HCARS, hyperspectral coherent anti-stokes Raman scattering; STELDI-MS, sorptive tape-like extraction coupled with laser desorption ionization mass spectrometry.

A similar study using ^1^H NMR and real-time polymerase chain reaction (RT-PCR) techniques determined transcript and metabolic profiles during the last four weeks of flowering. The additional metabolites identified were cannabichromenic acid (CBCA), inositol, acetic acid, fumaric acid, succinic acid, and choline. Partial least squares discriminant analysis (PLSDA) was used to classify metabolites. RT-PCR helped to monitor the expression levels of mRNA that encode the enzymes THCA synthase and CBDA synthase. Similar patterns of mRNA-encoding pathway enzymes and their respective metabolic product cannabinoids (THCA and CBDA) were observed [37]. Happyana and Kayser (2014) [18] also studied metabolite profiles in various plant anatomical structures including the trichomes, flowers, and leaves of both Bedrocan and Bedica cultivars. The concentration of THCA in the chloroform extract provided the most discriminating information among the cultivars tested. Thirteen compounds were identified in water extracts. Interestingly, asparagine was absent in the water extracts of Bedica trichomes and the presence of asparagine could be used to effectively discriminate among the cultivars. However, RT-PCR confirmed that THCA synthase was more expressed in leaves than in trichomes. These findings suggest that the expression of olivetolic acid synthase and olivetolic acid cyclase in trichomes triggers olivetolic acid production, which leads to THCA biosynthesis [18].

Proton NMR experiments were performed in dimethyl sulfoxide (DMSO)-D_6_ without internal standards. The total THC (THC_tot_) or the sum of THC, THCA, and CBN was used as a marker for cannabis extract potency. The ratio of THC_tot_/[CBD(A) + CBG(A)] was identified as a marker for chemotype, and the ratio of acidic/neutral cannabinoids reflected decarboxylation, which reflected extract quality. The ratio of (total cannabinoids/total phenolics, CAN_tot_/TPC) indicated the polarity of the extract. Selected NMR resonances of aliphatic (0–5 ppm) and aromatic (6–8 ppm) protons were used for the distinction of CBD-, THC-, and CBG-type cannabis. Cannabis extracts were prepared using solvents with a range of polarities to selectively fractionate compounds. The extracts reflected solvent polarity, where ethyl acetate recovered mostly cannabinoids, a 40% ethanol extract had moderate cannabinoids along with polar compounds, and a 70% methanol extract of heptane-defatted material was low in cannabinoids concentration. The authors also developed two HPLC/DAD methods as complementary tools that could differentiate chemotypes and determine extract polarity. This approach enabled the quantification of cannabinoids/acid derivatives (THC, CBD, CBG, and CBN) and flavones (homoorientin, orientin, isovitexin, vitexin, quercetin, apigenin, cannaflavin-A, and cannaflavin-B). In addition, phenol carbonic acids including chlorogenic acid were also identified [19].

Proton NMR was applied along with chemometrics approaches to differentiate cannabis extracts [38]. Cannabis samples were directly extracted in deuterated chloroform. ^1^H NMR and COSY were used to discriminate among cultivars with spectral ranges of 0.5–7.2 and 7.4–13.0 ppm, thus avoiding the resonance of chloroform. The linear discriminant analysis (LDA) provided the best prediction accuracy of 99.8 ± 0.4% for spectral profiling, and support vector classification machine trees (SVMTree) provided a robust tool and classification performance for ^1^H NMR spectra. ^1^D NMR has better reproducibility and an improved signal-to-noise ratio as compared to COSY. Tree-based classifiers used in multivariate analysis reduce non-linear classifications by dividing and conquering them into sets of smaller linear classifications. The large separations occur at the root of the tree and become more precise at the leaves.

### 2.2. GC-FID/MS

Analyses based on GC coupled with either MS or FID were used in metabolomic approaches to quantify and identify cannabinoids and terpenes. Subsequent multivariate data analysis can then be used to classify cannabis plants by their chemical diversity [20]. In Hazekamp’s group study [20], the compositions of cannabis and hemp accessions from the Netherlands were characterized. Eight major neutral cannabinoids and 36 terpenes were identified by GC-FID. Samples were extracted with absolute ethanol, and 1-octanol was used as an internal standard. All the acidic cannabinoids were fully converted to neutral cannabinoids at operating GC-FID detector temperatures (250 °C). A similar study was conducted on the composition and variability of cannabinoids, monoterpenoids, and sesquiterpenoids in 11 accessions grown in the same environmental conditions. In total, 9 cannabinoids and 27 terpenoids were quantified and PCA was used to discriminate among cannabis accessions. Higher levels of cannabinoids correlated with higher levels of terpenoids. Moreover, monoterpenoids can help to distinguish accessions containing similar cannabinoid and sesquiterpenoid profiles. The cannabinoid and terpenoid concentrations are reproducible for cannabis clones grown at separate times under standardized environmental conditions [21].

A GC-MS-based metabolomic study of two accessions of cannabis seed (CAN1 and CAN2) from different environments was performed. A total of 236 untargeted metabolites were identified, and 43 metabolites were significantly different between the accessions. The differing metabolites included cannabinoids, terpenes, fatty acids, carbohydrates, amino acids, organic acids, sugars, carboxylic acid, polyphenols, and polyamines. The crushed seed was extracted with methanol (70 °C) and centrifuged with ribitol as an internal standard. The supernatant was mixed with chloroform–water, separated into two solvent phases, dried, and derivatized for GC-MS analysis. Finally, PCA was performed to discriminate metabolite profiles among two seed samples. A temperate cultivar selected from a high-altitude site (CAN2) had higher concentrations of cannabinoids, alkaloids, amino acids, and fatty acids than the control cultivar (CAN1) [22].

A method that combined sorptive extraction using a stir bar followed by thermal desorption into two-dimensional GC (GC × GC) coupled with time-of-flight mass spectrometry was developed to analyze cannabis metabolites. The extraction was performed by mixing cannabis flower samples with a solvent mixture (water/methanol/acetone) for 60 min at 50 °C in the presence of a stir bar coated with polydimethylsiloxane. The untargeted metabolic profiling using 2D GC and PCA analysis identified 754 metabolites that belong to different chemical classes such as monoterpenes, sesquiterpenes, hydrocarbons, cannabinoids, terpenoid alcohols, and fatty acids. Finally, 70 statistically significant analytes were selected for discrimination among cannabis subspecies [23].

A terpenoid profiling approach was developed that employed a static headspace sampler (SHS), followed by GC-MS/MS, to quantify 93 terpenoids in 16 cannabis chemovars. Ground samples were subjected to direct measurement using an SHS, and chromatographic separations were conducted using a semi-polar stationary phase GC column. The selectivity for the quantification of overlapping compounds and increased sensitivity was achieved by the selected reaction monitoring mode in MS/MS experiments. The sample preparation methods (decarboxylation, isobutanol/ethanol/supercritical CO_2_ extraction) significantly impacted volatile terpenoid concentrations compared to untreated cannabis samples [24].

In a similar study, terpene metabolite compositions were compared for 33 chemovars using headspace GC-MS. A total of 67 terpenes were detected, including 29 monoterpenes and 38 sesquiterpenes. PCA analysis was performed to evaluate multivariate correlations and clustering among the metabolites. Nine major terpenes were present in the THC chemovars; however, three monoterpenes and four sesquiterpenes were predominant in CBD chemovars [25].

### 2.3. LC-MS

Berman’s group [26] used ESI-LC/MS to identify 94 cannabinoids from 10 different subclasses and compared 36 cannabis samples. The cannabis flower samples were extracted with ethanol, producing fractions that were then separated by HPLC for MS/MS analysis. LC-MS normalized data were distinguished according to the hierarchical clustering of cannabinoids in cannabis samples. The variation observed among cannabis samples was associated with CBD, THC-type chemovars, and decomposition products. Based on available analytical standards 13 cannabinoids were quantified while the remaining cannabinoids were identified based on masses obtained from the literature. The alkyl homologs elute from a reversed-phase column in the order C1-C3-C4-C5 (increasing lipophilicity). They also demonstrated that, despite the similar CBD content in the cannabis extract, the anticonvulsant effect of each extract differed. This finding elucidates the importance of the quantification of all cannabinoids [26].

An LC-based targeted metabolomics approach coupled with an untargeted analysis was used to study 11 known and 21 uncharacterized cannabinoids. Cannabis samples were extracted with 80% ethanol, then injected onto UHPLC and quantified against known standards. Cannabis strains were clustered into 5 distinct groups based on the total THC/CBD content in 33 commercial products. PCA and multiple linear regression were used to discriminate among the strains. Six unknown metabolites were unique to CBD-rich strains and three unknowns to THC-rich strains [27].

An LC-MS/MS was developed to identify six major cannabinoids, and LC-QTOF was designed to identify and fingerprint the seven minor cannabinoids in thirty cannabis samples. Cannabis samples were extracted using supercritical CO_2_ with ethanol as a cosolvent. Atmospheric pressure chemical ionization and multiple reaction monitoring for MS acquisition were used to quantify cannabinoids. Analysis of the LC-MS/MS data by PCA could discriminate among the varieties. The resulting data show differences in cannabinoids for plants grown indoors and outdoors. Specifically, higher concentrations of CBD, CBN, and THC were observed in outdoor-grown plants [28].

An untargeted metabolomics approach was used to discriminate among metabolites in cannabis extracts using LC-HRMS/MS and multivariate analysis. The chemical composition of cannabis samples extracted with ethanol and olive oil over time was compared. The major cannabinoids quantified include CBD, CBDA, THC, THCA, cannabigerolic acid (CBGA), and CBN. The other metabolites include trigonelline, proline, arginine, and choline. The cannabinoid concentrations were higher in ethanol as compared to olive oil extracts, while secondary metabolites predominated in olive oil extracts. The ratio of acidic to neutral cannabinoids was a discriminating feature present in both solvents [29].

Using GC-TOF/MS and LC-QTOF MS/MS in high-resolution mode, an untargeted analysis of polar and non-polar cannabis extracts identified 169 metabolites, with 134 in the non-polar extracts and 46 in the polar extracts. The non-polar hexane extracts include neutral cannabinoids, terpenoids, lipids, hydrocarbons, and benzenoids, and the polar methanol extracts include cannabinoids, amino acids, flavonoids, and carbohydrates. The composition of cannabinoid and terpenoid products differed for the same cultivars grown in the greenhouse vs. the field [30].

An LC-HRMS-based metabolic study was designed to determine secondary metabolite changes induced by exposing leaves to UV-C treatment. Powdered frozen leaf samples were extracted with isopropanol, filtered through SPE C18 columns, and subjected to LC-TOFMS analysis. LCMS data were recorded in both positive and negative electron ionization modes to obtain the *m/z* ratio, retention time, and area. Multivariate analysis was performed by PCA and OPLS-DA (orthogonal projections to latent structures discriminant analysis) to discriminate and highlight the important features. Changes in cinnamic acid amides and stilbene-related compounds were observed, but not for cannabinoid content [31].

The chemical profiling of dried commercial medical cannabis extracts was conducted for both the pre- and post-decarboxylation treatments. Dried cannabis was extracted with supercritical fluid (liquid carbon dioxide and ethanol as cosolvent) at ambient temperature to obtain a native extract. Decarboxylated products were prepared by heating the native extract to 170 °C. Ultra-performance liquid chromatography (UPLC) in both electrospray ionization (ESI) (+ve) and (−ve) modes was used to analyze both extracts. A total of 62 compounds were identified, including 23 phytocannabinoids, fatty acids, flavonoids, hydrocarbons, phenols, terpenoids, and other miscellaneous compounds. Not all compounds predicted from the heating of acidic cannabinoids or cannabinoid esters were present in the decarboxylated extract. Up to 26 predicted decarboxylation products were not detected [32].

### 2.4. Other Analytical Techniques

Additional analytical approaches have been employed for the spectral fingerprinting of cannabis samples. For example, thermal desorption ion mobility spectrometry (TD-IMS) was used to identify cannabinoids and discriminate different cannabis chemotypes. Powdered cannabis material was extracted with hexane, centrifuged, and subjected to TD-IMS analysis, where compounds were ionized using a 63Ni source. PCA, along with LDA, was used to cluster data and conduct chemotaxonomic discrimination [33].

Fischedick et al. (2009) [34] developed a rapid TLC system to quantify THC in cannabis samples. This system enables the qualitative analysis of neutral cannabinoids such as CBD, THC, ∆^9^-tetrahydrocannabivarin (∆^9^-THCV), CBG, and CBC. The use of normal-phase high-performance TLC plates with an automatic spotter and scanner provides a low-cost, high-throughput alternative for the forensic analysis and quality control of samples. The accuracy of this approach was confirmed by comparing results with those of a validated HPLC analysis. However, TLC has limited sensitivity and specificity compared to other methods [14,34].

Raman spectroscopy has also been utilized for the label-free, non-destructive, and chemically selective imaging of native biological samples. Hyperspectral coherent anti-stokes Raman scattering (HCARS) was used to identify and localize THCA or CBDA and myrcene in the secretory cavities of drug-type and fiber-type glandular trichomes, respectively. A spectral fingerprint that indicated the presence of CBGA was only found in drug-type trichomes. Two-photon fluorescence spectroscopy was also utilized along with HCARS to differentiate chlorophyll A from chloroplasts and organic fluorescence from cell walls [35].

Another study utilized sorptive tape-like extraction coupled with laser desorption ionization mass spectrometry (STELDI-MS). Cannabis samples were extracted using water–methanol solvents and then applied by spotting onto a silica gel 60 plate, which was subjected to chromatographic separation and MS analysis. This technique produced less signal suppression and no matrix–analyte adducts were formed. Therefore, the approach was an improvement over MALDI without a normal phase separation step. The major cannabinoids detected were CBD, CBN, THC, CBC, CBDB, cannabichromevarinic acid (CBCVA), and cannabidivarinic acid (CBDVA). Moreover, markers associated with preservatives used in processing, such as ethyl 4-hydroxybenzoate, propyl 4-hydroxybenzoate, and butyl 4-hydroxybenzoate, were identified [36].

## 3. Current System and Potential Quality Issues

### 3.1. Current System

Currently, the following four systems play an important role in maintaining the safety testing and quality of cannabis (Figure 2).

#### 3.1.1. Analysis for the Enforcement of Government Regulations

Cannabis is a complex mixture containing various bioactive compounds. Government regulatory agencies in countries where cannabis is legal enforce minimum testing requirements for maintaining the safety and quality of these products. For example, the Canadian regulatory agency Health Canada has testing requirements for cannabinoids (THC, THCA, CBD, CBDA), microbials (mold, yeast, bacteria, mycotoxins, aflatoxins), chemical contaminants (residual solvents), pesticides, and heavy metals (arsenic, mercury, lead, cadmium). However, this system has drawbacks in covering all the compounds, as the same cultivar from different locations can have different concentrations based on its growing conditions and processing. The regulations have some limitations, including consistent analytical testing and defining cannabis categories; potency limits and variability between industries, accurate vaping technologies, and individual packages relative to labeling; and quality assurance in dispensing products [39]. In another instance, the US 2018 farm bill regulations allow hemp cultivation with 0.3% THC (on a dry weight basis) but do not mention other hemp-derived cannabinoids such as ∆^8^-THC. This forms a risky situation where the product is sold to individuals of all ages in some US states [40,41]. Hence, it is important to have scientific evidence-based regulations and testing regimens for all compounds to ensure consumer safety and product quality.

#### 3.1.2. Industry/Pharmacy/Retail Run

In the industry model, there are no standardized testing methods and no guidance provided by regulatory agencies. Hence, industry must develop its own methods, create standardized operating procedures (SOP), follow SOPs, and test only a limited set of compounds as required by regulators. Samples are also tested by third-party labs. However, this model also has drawbacks, as different labs use different analytical procedures, instrumental methods, and calibration standards, leading to variation in results [42]. Normally, companies do not share their analytical procedures with one another due to proprietary issues. There are also instances where THC inflation has been reported by specific labs to profit from their partners [43].

#### 3.1.3. Consumer Led

Consumers are more prone to buying cannabis based on the THC content and visual and sensory evaluation as quality indicators for recreational use [44]. It was speculated that high THC concentration will give a more desired effect; however, studies have shown that the effect is not based on the potency and is more complex. In one of the studies with 121 participants, half of them were provided with very high THC extracts (70% or 90% THC) and another half with cannabis flower (16% or 24% THC). However, it was found that the neurobehavior patterns were similar for both groups [45]. Hence, it is necessary to provide consumer education about quality, and they should be involved in future regulation systems.

#### 3.1.4. Law Enforcement Agencies

Enforcement agencies also play an important role in cannabis regulation and enforcement across borders. However, current systems/forensic labs are majorly focused on THC toxication in biological samples (breath, blood, urine) [46]. As a number of synthetic analogs have been developed, and with the natural variability of cannabis, these tests are unable to detect other psychoactive compounds.

### 3.2. Lack of Standardized Extraction and Refinement Methods

Different extraction methods have been used to prepare cannabis extracts. The quality and composition of any plant extract are highly dependent on the extraction process. Extraction solvents range from water, hydrocarbons (butane, pentane, hexane), alcohols (ethanol, isopropanol), supercritical carbon dioxide, and many blends of these extractants. Cannabis compounds vary in polarity, molecular weight, and other properties that affect solubility in solvents that range in polarity. Extraction with water, a polar solvent, can be used to recover the entire trichome. Ethanol, with intermediate polarity, can recover flavonoids and pigments. Low-polarity solvents such as hydrocarbons do not dissolve chlorophyll and water. Supercritical carbon dioxide and mixed solvents can extract a wide range of compounds with similar low polarity. Once cannabinoids and associated molecules are extracted, they can be enriched and refined using short-path distillation, wiped film molecular distillation, and winterization [47]. These different processes produce varied chemical compositions in final products. For example, cannabis extracts are labeled based on the cannabinoid content, but due to the different extraction techniques involved, the overall concentration of other chemical compounds is altered and, in turn, changes the biological activity of the cannabis extract.

### 3.3. Cannabinoid and Terpenoid Stability

Cannabinoids and terpenoids represent the major bioactive component in cannabis and are biosynthesized by specific enzymes. The overall chemical composition and concentration of these compounds differ by the plant’s genetics, age, growing conditions, stage of maturity at harvest, drying, storage, extraction, and formulation methods. However, these compounds easily degrade during post-harvest storage and the activation of acidic cannabinoids. Plant genetics plays an important role in preserving the bioactive compound profile. By growing in a greenhouse with controlled conditions, a consistent chemical profile can be produced. Moreover, slight variations can significantly affect the ratio and synergistic effects among different compounds and can affect overall activity [48]. Furthermore, if the cannabis flower is exposed to air and light for a prolonged period (not a controlled environment), acidic cannabinoids such as THC-A are oxidized to cannabinolic acid (CBN-A) and further converted to cannabinol (CBN). Moreover, CBN is reported as a weak psychoactive cannabinoid with mostly mild analgesic and anticonvulsant activity. Similarly, CBN can also be formed during the decarboxylation of THCA to THC [49]. On the other hand, terpenoids are volatile compounds, and storage (temperature and time) greatly affects their concentration. Terpenoids may also decompose via oxidation, isomerization, polymerization, thermal rearrangement, and dehydrogenation [50]. Concentration variations and byproduct formation have a negative effect on product quality and safety. Hence, the exact concentration of the cannabinoids and terpenes present in a food product must be disclosed (labeled), their stability enhanced (antioxidants), and the dosage guaranteed until expiry.

### 3.4. Lipid Oxidation Products

Cannabis extracts are often sold in vegetable oil carriers. There are reports of the interaction of lipid oxidation products with cannabis extracts. The oxidation products include oxygenated terpenoids such as verbenol, linalool, alpha-terpineol, terpinen-4-ol, aldehyde, alcohols, and ketones. Natural terpenes can undergo photo-oxidation in the presence of light and singlet oxygen. The first products formed are unstable allylic hydroperoxides. The spontaneous rearrangement of these oxidized products produces alcohols that are often further oxidized to their respective aldehydes and ketones. For example, limonene degrades to *trans*- and *cis*-metha-2,8-dien-1-ol and *trans*- and *cis*-carveol during photo-oxidation [51,52]. These cannabis oils are activated by heating, which decarboxylates acidic cannabinoids to produce neutral cannabinoids. This heating mediates the formation of several “ex novo” lipid breakdown products such as ketones and aldehydes and can significantly influence oil digestibility and stability. It was observed that the concentration of ketones and aldehydes was lower under refrigerated conditions compared to room temperatures. Headspace-solid-phase microextraction (HS-SPME) coupled with GC-MS was used to profile volatile compounds to understand storage (6 weeks) and temperature’s effects on cannabis-containing oils [52]. The formation of lipid oxidation products for cannabis macerated oils mostly depends on extraction method temperature, fatty acid composition (oil matrix of cannabis extract), and storage temperature. For example, medium-chain triglyceride oil is less susceptible to degradation compared to olive and hemp seed oils [53].

### 3.5. Structural and Stereoisomers

There are several structural and stereoisomers reported for cannabinoids. Cannabinoids are made up of three moieties the isoprenyl residue, the resorcinyl core, and the sidechain [54]. THC can theoretically exist as seven structural isomers, Δ^6a,10a^-THC, Δ^6a,7^-THC, Δ^7^-THC, Δ^8^-THC, Δ^10^-THC, and Δ^9,11^-THC. These isomers have the same molecular formula but different bonding arrangements around the double bond from C9-C10 across the terpene ring. Currently, no analytical method is available for these isomers except Δ^8^-THC; the total THC content is calculated as combinations of Δ^8^-THC, Δ^9^-THC, and THCA. THC has two stereocenters and occurs as four stereoisomers: (–) *trans*, (+)-*trans*, (–)-*cis*, (+) *cis*. Only (–) trans-Δ^9^-THC occurs naturally in the plant in the form of Δ^9^-THCA. Similarly, CBD occurs as two stereoisomers (–) CBD and (+) CBD, in which (–) enantiomer is a naturally occurring compound. These stereoisomers can be separated using chiral chromatography [55,56].

### 3.6. Adulteration

Cannabis has been modified or diluted with different types of adulterants. Adulteration might be performed to extend cannabis extracts with materials for economic gain, enhance the efficacy of low-quality cannabis, or mitigate cannabis’ side effects. Cannabis has also been mixed with synthetic analogs (cannabimimetics) termed Spice/K2 products. These analogs were initially synthesized to study the endocannabinoid system and develop therapeutically effective compounds. However, they have become subject to drug abuse. The number of these analogs is growing, and if some of them became regulated, they were replaced by another analog in the market to satisfy demand. LC-MS, GC-MS, and direct analysis in real-time (DART)-MS have been used to identify and measure these compounds, but the limited information regarding their chromatographic/spectral information provides some challenges [57]. Adulterating cannabis with tobacco, calamus, or other cholinergic agents can increase the effects of cannabis or reduce adverse effects. There was a report of admixtures of cannabis and calamus root to reduce the adverse effects of cannabis. It was reported that beta-asarone in calamus roots blocks acetylcholinesterase which diminishes cannabimimetic effects [58]. CBD can convert into THC in an acidic environment under laboratory conditions. In addition, terpenes can be converted to toxic degradants such as benzene (carcinogen) and methacrolein [55]. There were also reports of the adulteration of cannabis oil using pine rosin, NMR, and ESI-MS to identify pine rosin ingredients such as abietic and other resin acids. This can lead to inhalation toxicity in e-cigarettes and vaping products [59]. Similarly, vitamin E acetate was used as a major diluent in illicit cannabis vaporizer cartridges, detected by untargeted analysis (GC-MS and LC-MS/MS) [60].

## 4. Authentomics

Authentomics analysis can provide verified analysis as a food screener that can protect consumers from fraud. Figure 3 summarizes the schematic of authentomics.

### 4.1. Lessons from the Food Industry

In the last few years, there has been growing interest from consumers, producers, and food authorities for agro-food product quality and safety through the food chain from farm to fork. There are instances of fraudulent acts motivated by economic returns. Typical fraud can include dilution, tampering, adulteration, or the misrepresentation of food, food ingredients, or food packaging. Examples include spirit adulteration with methanol in place of ethanol. Such action led to the death of 38 people in the Czech Republic [61]. Similarly, in China, milk was adulterated with melamine to increase the nitrogen content. Nitrogen content measurements have been used in official methods as a surrogate for protein content. The unscrupulous addition of a high nitrogen content compound such as melamine to food prior to testing would indicate a higher nitrogen content and consequently be falsely interpreted as a higher protein in the product [62]. Food and food byproducts are made up of a complex matrix of various compounds/metabolites in different concentrations. They are generated from biological materials handled through supply chains that can be complex, involving cultivation, storage, shipping, processing, packaging, and distribution. An authentic food product is “*a food product where there is a match between the actual food product characteristics and the corresponding food product claims; when the food product actually is what the claim says that is*” [63,64]. In a discussion paper on food integrity and food authenticity from the working group of the Codex Alimentarius Commission, food authenticity is described as “*…the quality of a food to be genuine and undisputed in its nature, origin, identity, and claims, and to meet expected properties*”, food fraud is described as “*any deliberate action of businesses or individuals to deceive others in regard to the integrity of food to gain undue advantage*”, and food integrity is described as “*The status of a food product where it is authentic and not altered or modified with respect to expected characteristics including, safety, quality, and nutrition*” [65]. The authentomics approach has been applied in the analysis of foods including wine, honey, juice, beer, olive oil, milk, and coffee. Food products are considered authentic if manufactured with proper quality procedures and all the chemical components are consistent. With proper processing, food should have a reproducible chemical composition. Food authentication has attracted interest among shareholders such as food producers, importers, exporters, consumers, regulatory agencies, law enforcement, and the scientific community. Hence, a comprehensive approach is necessary to characterize the molecular constituents of food. Authentomics is not only related to product quality but also safety and health. Hence, rapid and robust analytical methods, reliable biomarkers, and big data analysis are important tools to overcome food fraud. There are strict regulations for food safety and authenticity across the world. The US FDA has developed a food safety modernization act (FSMA) to ensure a safe food supply by preventing contamination [66]. Similarly, the European food safety authority (EFSA) evaluates the risk associated with the food chain [67].

The knowledge of food authentomics could improve the analysis of cannabis products in several ways. Firstly, authentomics involves a comprehensive, non-targeted analysis of a food product, considering all its molecular constituents. This approach could be applied to cannabis products to provide a more complete picture of their composition, including the concentration of bioactive metabolites and any contaminants. This could help regulators and researchers to better understand and regulate the complex chemistry of cannabis products and to assure their safety and efficacy. Secondly, the use of reliable biomarkers and big data analysis could help to verify the authenticity of commercialized cannabis products by comparing their molecular profiles to those of historically authenticated samples. This could be particularly important in the cannabis industry, where there have been instances of fraudulent activities motivated by economic returns. Overall, the application of authentomics to the analysis of cannabis products could provide a more robust and reliable approach to ensuring product quality, safety, and authenticity and could help to protect consumers from the potential harm of counterfeit products.

In the case of cannabis products, the application of authentomics knowledge can also play a crucial role in improving the analysis and understanding of the hedonic qualities of these products. The authenticity of the chemical composition of the product can significantly impact sensory and hedonic attributes such as aroma, flavor, and potency. For instance, the presence of contaminants or unauthorized additives can lead to negative effects on the sensory experience of the product. On the other hand, the ability to accurately determine the chemical composition of the product can also help to understand the relationship between its chemical constituents and the sensory and hedonic qualities. This knowledge can assist in developing consistent and high-quality cannabis products that meet the expectations and preferences of consumers.

### 4.2. Identification–Conformity–Quantification

One of the challenges in the food industry is to detect unexpected adulterants. For example, it was unexpected that melamine would be used as an adulterant in milk [62]. Hence, it is better to develop techniques to analyze both known and unknown (novel) compounds in the food components through targeted and untargeted analyses, respectively. In recent years, progress in analytical techniques has improved food authenticity and traceability. Some techniques include liquid and gas chromatography coupled with mass analyzers, DNA-based techniques, sensor techniques (electronic tongues, electronic noses), and other spectroscopic techniques (NMR, vibrational, fluorescence). Spectroscopic techniques can provide non-destructive platforms for non-invasive analyses that are rapid, easy to operate, and can be applied both for routine analysis and in food control laboratories in the industry [68]. The two primary analysis approaches of targeted and non-targeted analyses will prove important in the characterization of cannabis products.

#### 4.2.1. Targeted Analyses

Targeted analyses (TA) are used where metabolites are known, or specific biomarker compounds can be used to assess the purity of authentic food. For targeted analysis, it is important to have analytical procedures validated. Some examples include anthocyanin derivative analysis to determine red wine age using an HPLC-MS/MS and CIELAB approach for 234 different vintages of red wine. Red wine aging is closely related to changes in anthocyanin composition and chromatic characteristics, regardless of environmental factors, variety, and winemaking technique. The anthocyanin stabilities were: pinotins > flavanyl-pyranoanthocyanins, vitisin A > monomeric anthocyanin, direct anthocyanin-flavan-3-ols condensation products > vitisin B, anthocyanin ethyl-linked flavan-3-ols products. Vitisin A, pinotin, and flavanyl-pyranoanthocyanins contributed significantly to each wine’s prolonged aging color [69]. In another example of targeted analysis, polyphenol compounds in seeds such as flax, chia, and sesame were used as markers for authenticity in bakery products. Polyphenols were analyzed by HPLC-DAD-ESI-qTOF (MS/MS) in different seeds and a chemometric approach was used to classify 12 compounds that acted as markers for discrimination among samples. The concentration and presence of lignans and hydroxycinnamic acid allowed discrimination among groups. The proposed markers were stable during baking and could be used to authenticate bakery products and raw materials containing these seeds [70].

Targeted analyses of cannabis and food products are mostly similar. The analysis of either matrix involves identifying specific compounds of interest and measuring their concentration. In both cases, the goal is to provide accurate information regarding product composition. The methods used to analyze the compounds of interest in food products and cannabis products can differ, as the latter may require more specialized techniques due to the presence of psychoactive compounds with unique properties. Despite these differences, the principles of targeted analysis are the same, as they provide a detailed and accurate picture of product composition.

#### 4.2.2. Non-Targeted Analysis

The analysis of food ingredients can produce chemical fingerprints. The chemical composition (fingerprint) is an excellent indicator of origin, authenticity, quality, and/or adulteration. Fingerprint variations may indicate changes in the metabolite levels caused by different factors including the geographical origin of the raw materials, production systems, adulteration, or storage conditions. A database of the fingerprint data of known food products is an essential tool in determining authenticity. After the database is developed, the authenticity of food can be affirmed by comparing a fingerprint of that food with an authentic food fingerprint from the database. These chemical fingerprints are obtained using various analytical technologies, which are selected based on the food and attributes. For example, a non-targeted analysis of virgin olive oils (extra virgin olive oil, EVOO; virgin olive oil, VOO; lampante olive oil, LOO) was performed by flash GC. A training set of 331 representative samples was collected, representing different harvesting processes, geographical origins, sensory attributes, and olive oil cultivars, and was analyzed. The raw data collected from GC fingerprinting of the volatile fractions were interpreted using multivariate analysis (PLSDA). This approach provided a superior alternative to sensory panels, increasing efficiency and rapid screening for the classification of olive oil by quality [71]. This type of analysis can be applied in any laboratory or industry as a quality control measure. Another example includes the application of Fourier transform near-infrared spectroscopy (FT-NIR) along with chemometrics to discriminate between white truffles *Tuber borchii* and *T. magnatum* and black truffles *T. aestivum*, *T. indicum*, and *T. melanosporum*. These truffles are sold at a high price due to the unique aroma and taste emitted from the fruiting bodies. For example, *T. magnatum* price ranges between 3000 and 5000 €/kg, and *T. melanosporum* costs 700–1200 €/kg. The large price difference increases the chances of the fraudulent misrepresentation of species of similar morphological appearance. Lyophilized truffle samples are rich in amino acids and dietary fiber. The selective bands observed for proteins and amino acids are at 6667 cm^−1^ for N-H stretching bands. Furthermore, N-H bands at 4859 cm^−1^ and 4600 cm^−1^ were observed for amide groups. In this study, 75 samples from different geographical origins and harvest years were analyzed using FT-NIR. PCA discrimination afforded greater than 99% classification accuracy. In addition, an accuracy of >83% was achieved for differentiation between Italian and non-Italian *T. magnatum* samples. FT-NIR provided a simple, cost-effective, reliable, easy-to-handle solution to discriminate and authenticate truffle species [72].

The non-targeted analysis of cannabis and food products share similarities in that this approach is designed to understand a sample’s chemical composition comprehensively. A non-targeted analysis does not focus on pre-selected compounds and instead quantifies all measurable compounds present. This approach provides a more complete picture of the sample by detecting both known and unknown compounds.

However, there are also differences in the non-targeted analysis of cannabis products and food products. The complexity of the chemical composition of cannabis products is often more significant compared to food products, as they contain numerous compounds, including cannabinoids, terpenes, flavonoids, and residual solvents. This complexity can make the non-targeted analysis of cannabis products more challenging than food products. Additionally, the regulations and legal considerations surrounding the analysis of cannabis products are different from food products, and as a result, the methods used for their analysis may also differ.

### 4.3. Food Metabolome Database

There are several public-domain food metabolome databases available for comparisons with known and unknown metabolites present in food and food components. These databases are important tools for biomarker discovery, clinical chemistry, metabolomics, and general education. Some examples include the human metabolome database containing 114,265 human metabolites. The human metabolome database includes compounds found in common foods, as these are present in the human body prior to metabolism [73]. The food database provides information on macronutrients and micronutrients, including compounds that contribute to food color, flavor, texture, and aroma. There are >28,000 metabolites reported in the database. Included is information regarding compound nomenclature, structure, chemical class, physico-chemical data, food source, and concentration in various foods [74]. Another online database is Phytohub, which provides information regarding dietary phytochemicals and their human and animal metabolites. It includes secondary plant metabolites such as polyphenols, terpenoids, and alkaloids and is designed to be used in nutritional metabolomics. It includes other information such as food source, molecular formula, monoisotopic mass, and MS/MS fragments [75]. The Phenol-Explorer database explores foods’ polyphenol content. The database contains > 35,000 content values for 500 different polyphenols in over 400 foods. The polyphenol data before and after processing were collected from peer-reviewed publications. The major data belong to fruit and vegetable food groups and their polyphenolic compounds. Cereals and oils are poorly represented in the database [76]. The yeast metabolome, which consists of the metabolite found in or produced by Saccharomyces cerevisiae (also known as Baker’s yeast or Brewer’s yeast), is also available as a database. This database is useful for the study of the origin and fate of yeast metabolites in a number of food products such as wine, bread, and beer, which are produced by yeast fermentation [77]. These metabolome databases are important for identifying different metabolites in the food matrix. Initially, the profile of the chemical fingerprinting of representative samples should be created. The profile can then be compared with a large database of spectra for known authentic samples. For example, “Metabolights” [78] is an open-access online repository where spectral, structural, and chromatographic data are shared for cross-species and cross-technique analysis. The database also contains information regarding the biological roles, concentration, origin, and metabolic pathways of metabolites. In some instances, raw experimental data from different experiments are included. The user studies in the database are labeled with a unique identifier for publication reference [79,80]. These raw data can potentially be used for external validation and aid in the comparison of results from different laboratories and identify robust markers for detecting food fraud. This can help with quality control monitoring and testing for determining purity and authenticity.

The development of a comprehensive cannabis metabolome database and standard methods that approach the detail and quality used in the authentomics analysis of food products is essential for ensuring the quality, safety, and efficacy of cannabis products. As the legal landscape for cannabis products continues to evolve, it is important to have robust analytical methods in place that can accurately and reliably assess the composition of these products. The use of a cannabis metabolome database and standardized methods will allow for the consistent and reproducible characterization of the molecular constituents of cannabis, which will be crucial for both regulatory compliance and consumer confidence in the safety and quality of these products. By incorporating the advances made in the authentomics analysis of food products, the cannabis industry can ensure that its products are of the highest quality and that consumers can have confidence in their safety and efficacy.

## 5. Future Directions

### 5.1. Non-Targeted Analysis

Cannabis is a complex chemical matrix composed of various types of secondary metabolites, including cannabinoids, terpenoids, flavonoids, and phenolic compounds. The plant is available in many different forms, including dried flower, pre-rolls, seeds, and vapes, extracts such as oils, capsules, resin, rosin, isolates, distillates, shatter, wax, hash, and kief, edibles such as chocolates, gummies, baked goods, confectionery, and beverages, and topicals such as creams, lotions, and bath salts. Currently, most targeted analysis methods for cannabis focus on quantifying specific compounds such as cannabinoids, terpenes, and contaminants such as residual solvents, pesticides, heavy metals, microbial contaminants, mold, bacteria, and yeast. However, only a limited number of non-targeted analysis methods have been developed, and they mostly lack validation and are performed in-house.

Non-targeted analysis provides a significant advantage over targeted analysis in terms of identifying novel compounds or contaminants. For instance, the non-targeted approach can help identify contaminants such as vitamin E acetate, which was not known before and was used as an adulterant in vape cartridges [60]. By providing a comprehensive view of the chemical composition of cannabis products, non-targeted analysis can help ensure their safety and quality for consumption. Thus, there is a growing need for a cannabis metabolome database and standard methods that are as detailed and high-quality as those used in the authentomics analysis of food products.

### 5.2. Standardized Analytical Procedure

There should be standardized analytical procedures for the analysis of compounds in cannabis extracts. Such analyses help to avoid analytical variations and can lead to using standard protocols across other labs. United States Pharmacopeia (USP) and the Association of Official Analytical Chemists (AOAC) have developed methods for cannabinoid analysis [81,82]. These standard methods will help industry partners and testing laboratories follow these standard procedures, validate their results, and use them for in-house testing. It will help to increase consumer confidence regarding labeling claims by the manufacturer.

### 5.3. Experimental Flow Design

A general experimental workflow is shown below (Figure 4).

The first important step is to design experiments involving a choice of samples to collect, the sample-handling procedure, detection methods, and expected outcomes. The second step requires sample handling with proper storage and temperature to mitigate the degradation of compounds. Sample preparation starts with harvesting (flower samples) and involves quenching, homogenization, and storage. For example, cannabinoids and terpenes are susceptible to heat and light and can easily convert to other byproducts. It is important to obtain representative samples of the batch to avoid any variation due to the heterogeneity of the cannabis matrix. Based on the detection methods, various solvents can be used for the extraction of metabolites. It is better to reduce the number of extraction steps to avoid any metabolite loss. This might require an individual/combination of polar and non-polar solvents to extract different metabolites from cannabis samples. One option is the application of deuterated solvents such as deuterated chloroform or methanol to extract prior to NMR analysis. Non-targeted methods generate very large amounts of data and require multivariate analysis tools such as PCA, PLSDA, and hierarchical cluster analysis (HCA) to discriminate among samples. It is useful to have access to a database to store information and share it among various stakeholders. Finally, an analytical method should be appropriately validated and tested among different laboratories for variations [83,84].

### 5.4. Open-Access Structure Databases

A chemical structure database containing metadata and spectral information is necessary to authenticate samples. The development of such a database requires open access to data sharing among various shareholders. The database must be clearly defined for its intended use and end-users. Implementing such databases is resource-intensive and includes defining the database scope, the collection of authentic and representative samples, sample preparation, data acquisition and validation, database storage, accessibility, and data validity. The supply chain risk assessment must be performed during database planning to determine the highest risk of fraud; for example, for dried herbal cannabis samples, the risk is at the geographical origin for the identification and traceability of chemovars, and for the finished products (edibles, concentrates) the metabolite profiling must be performed to determine any variability. The samples in the database should be authentic to be included [85].

### 5.5. Authentic Standards

An important consideration should be given to obtaining authentic standards for analysis. Due to the former stigma and the legal status of cannabis, research into its analysis has faced major roadblocks, especially in acquiring standards. There are 13 USP quality standards available for cannabinoids [81]. The availability of authentic standards ensures the availability of a foundation from which to ascertain the identity, purity, and potency of cannabis and reduces the chances of adulteration. As the number of known metabolites increases, more resources must be allotted to authenticate standards for identification and quantification. Standard purity and storage conditions should be emphasized, as these standards might be degraded by light and heat. Standards should be tested/accessed before analysis to confirm reproducible and accurate results from samples [86].

### 5.6. Public–Private Partnership

It is important that both public (government agencies, regulatory bodies, and academic institutions) and private (industry and testing labs) shareholders work together in developing and sharing data and experimental approaches. Most targeted and non-targeted analyses have been developed and validated “in-house”, with little effort devoted to inter-laboratory reproducibility. The sources of variation in the laboratory can occur at the sampling, sample preparation, instrumentation, and data mining/handling stages. This can be true for the analytical variation of different personnel in the same laboratory [17]. Emerald bioscience has developed a proficiency test program called “Inter-Laboratory Comparison and Proficiency Test (ILC/PT), The Emerald Test™” for cannabis and hemp testing. It is accredited by the International Organization for Standardization/International Electrotechnical Commission (ISO/IEC) 17043, which is a provider of proficiency tests. Samples are distributed to different labs, and the results are submitted through an electronic data portal. The individual labs receive results from both their own lab and other peer labs for comparison. Labs that perform within a specific tolerance in each proficiency test category established by the ISO provider receive the Emerald Badge™ [87].

### 5.7. Markers for the Standardization of Herbal Drugs and Extracts

Markers are essential for the identification of product variation and authentication and for determining quality. One of the markers used to reflect the age of cannabis products is CBN, which was formed by different pathways, such as the decarboxylation of CBNA, which originates from THCA oxidation or the oxidation of Δ^9^-THC. CBN is present in very low amounts in fresh tinctures (cannabis extract) or dry cannabis flowers. Other stability markers have also been suggested, such as THC_tot_ (THC +THCA + CBN) or CAN_tot_ (total acidic + total neutral cannabinoids), as well as ratio markers of THCA/THC, CBGA/CBG [49]. Moreover, α-terpinolene is a genetic marker that can distinguish between two gene pools for breeding low THC varieties and may be related to the geographical origin of cannabis materials [53,88].

### 5.8. Minor Cannabinoids and Their Pharmacology

There has been more focus given to THC and CBD due to their initial discovery and bioactivity, but cannabis contains more than 100 cannabinoids. It will be useful to understand the pharmacological activity of minor cannabinoids. These compounds should be synthesized (less concentration in cannabis plants) and tested for their biological activity. Most other cannabinoids such as CBDA, ∆^9^-THCV, CBDV, CBG, and CBC are non-intoxicating. Some have great potential in medicinal applications and will help in our understanding of interactions among the compounds present in cannabis [89,90].

### 5.9. Personalized Medicine and Pharmacometabolomics

The endocannabinoid system (ECS) of individuals varies considerably, leading to significant differences in the responses of individuals to cannabinoids. Progress in genomics might elucidate the role of genetic variability in response to cannabis. It is possible that genomic sequencing could help to understand differences in individual responses to cannabinoid therapy. Personalized approaches will be developed in the future that target therapies for individuals with specific ECS genetic variants and/or individuals expressing biomarkers.

## 6. Conclusions

This study summarized authentomics using metabolomics to confirm the authenticity of cannabis, which is effective in medical and health promotion. This study of cannabis’ big data analysis through authentomics demonstrates the quantitative analysis of commercially available cannabis product ingredients and patient samples. Mechanisms to certify cannabis ingredients are being developed, but performing these analyzes requires a global implementation of an auxiliary system that generates reliable data and verifies authenticity. The authentomics approach to food analytics, which is currently being pioneered, provides an international sentry system including infrastructure, an approved ISO method for numerous foods, and a software and hardware framework to apply authentomics to any food product. It is designed to augment or replace targeted analysis by applying this approach to cannabis authentomics. The future development of authentomics and targeted analytics services for cannabis could be coordinated with the world’s leading authentomics instrument and data analytics providers. Therefore, authentomics platform technology should be implemented into the cannabis quality system to meet the ISO standards verified by cannabis’ integrated database design and construction. This authentomics platform technology would meet the needs of the cannabis industry to provide a robust analysis of target substances while collecting information that would capture currently intangible aspects of cannabis chemistry. The combination of targeted and untargeted analysis is essential to monitor the complex chemistry of cannabis products. Untargeted approaches require the use of big data to capture variability that is otherwise dismissed as unknown. Cannabis analysis is, therefore, much like that of food, requiring empirical knowledge of the concentration of regulated components and more subjective knowledge that would relate to the experience of the consumer. Therefore, this review combines the inside knowledge of the cannabis industry with the latest applications of the non-targeted analysis of plant materials.

## Figures and Tables

**Figure 1 ijms-24-08202-f001:**
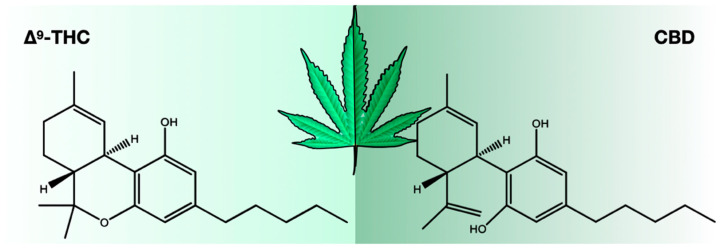
Chemical structures of Δ^9^-THC and CBD.

**Figure 2 ijms-24-08202-f002:**
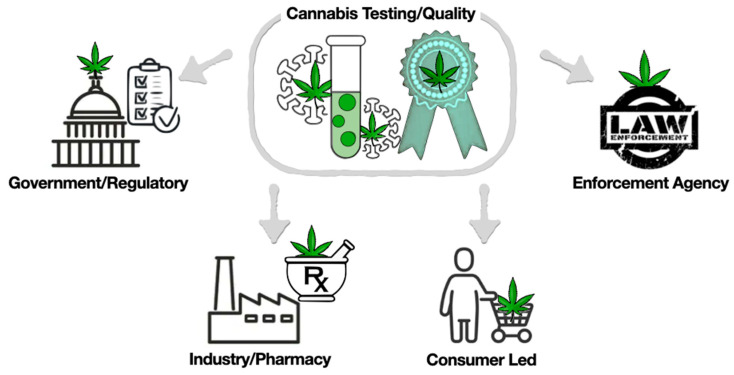
Four systems play an important role in maintaining the safety testing and quality of cannabis.

**Figure 3 ijms-24-08202-f003:**
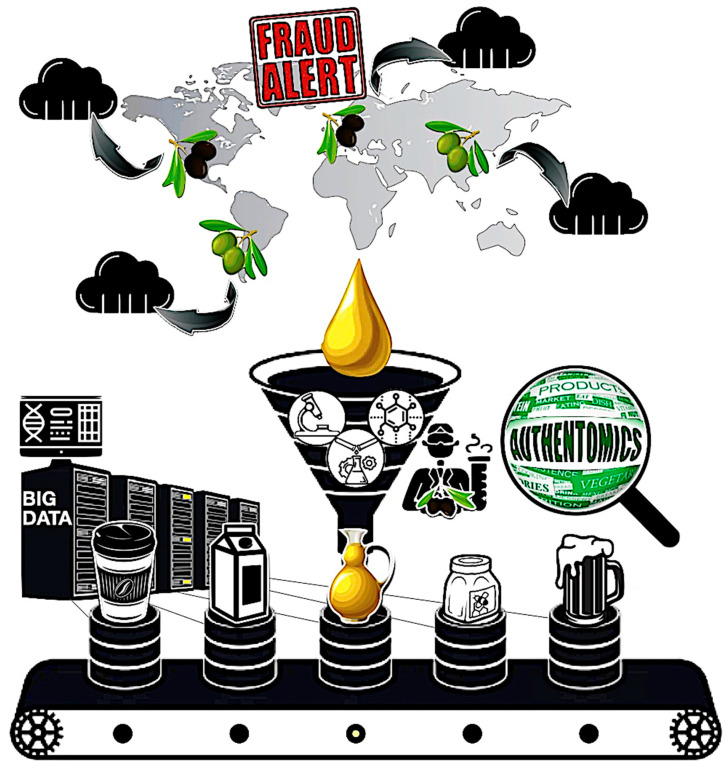
Schematic of authentomics as a food screener expecting a standardized platform.

**Figure 4 ijms-24-08202-f004:**
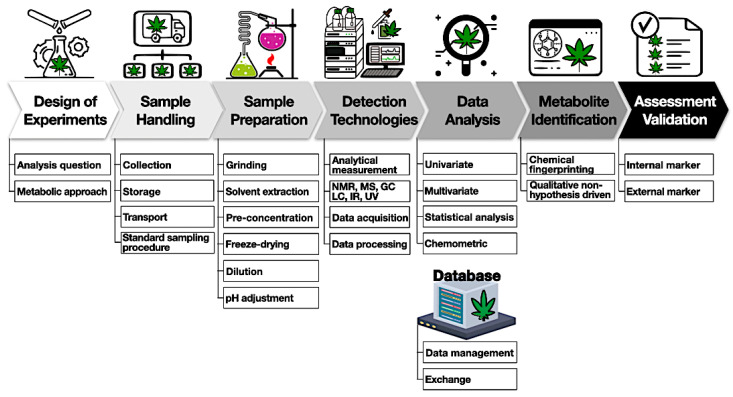
Cannabis’ general experimental workflow.

**Table 1 ijms-24-08202-t001:** Chemical constituents (metabolites) of *Cannabis sativa* L. (flowers) by chemical class.

Chemical Class	1980 ^a^	1995 ^b^	2005 ^c^	2014 ^d^	2017 ^e^
Cannabinoids type	61	66	70	104	120
CBC	4	4	5	8	9
CBD	7	7	7	8	7
CBG	6	6	7	17	16
Δ^8^-THC	2	2	2	2	5
Δ^9^-THC	9	9	9	18	23
CBE	5	5	5	5	5
CBL	3	3	3	3	3
CBN	6	7	7	10	11
CBND	2	2	2	2	2
CBT	6	9	9	9	9
Miscellaneous	11	12	14	22	30
Nitrogenous compounds	20	27	27	29	33
Amino acids	18	18	18	18	18
Proteins, enzymes, and glycoproteins	11	11	11	11	11
Sugars and related compounds	34	34	34	34	34
Hydrocarbons	50	50	50	50	50
Simple alcohols	7	7	7	7	7
Simple aldehydes	12	12	12	12	12
Simple ketones	13	13	13	13	13
Simple acids	20	20	20	20	20
Fatty acids	12	23	23	27	27
Simple esters and lactones	13	13	13	13	13
Steroids	11	11	11	15	15
Terpenes	103	120	120	120	120
Non-cannabinoid phenols	16	25	25	33	33
Flavonoids	19	21	23	27	27
Vitamins	1	1	1	1	1
Pigments	2	2	2	2	2
Elements	0	9	9	9	9
Total	423	483	489	545	565

Source Ref ^a^: [7]; Ref ^b^: [8]; Ref ^c^: [9]; Ref ^d^: [10]; Ref ^e^: [11]. Abbreviations: CBC, cannabichromene; CBD, cannabidiol; CBG, cannabigerol; THC, tetrahydrocannabinol; CBE, cannabielsoin; CBL, cannabicyclol; CBN, cannabinol; CBND, cannabinodiol; CBT, cannabitriol.

## Data Availability

The data of the current study are available from the corresponding author upon reasonable request.

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
