# Peer review of "A Metabolomics and Big Data Approach to Cannabis Authenticity (Authentomics)"

_ijms, 2023, doi:10.3390/ijms24098202_

Round 1

Reviewer 1 Report

Dear authors,

I read the paper with great interest. Despite you making a lot of effort in this review summarizing more than 80 up-to-date references, some modifications are required. The objective described in the title is not entirely addressed.

The chemistry of the Cannabis plant is very complex and some important aspects we're not considered; for example, in table 2, you undervalued which part of the plant has been used for the extraction, which cultivar, etc

Furthermore, there isn't any analysis or 'discussion' but only a lot of statements about what is in the literature with examples off-topic.

I think you need to rethink the intention of your manuscript and reorganize it better.

Author Response

Thank you for your patience and recommendations for strengthening our manuscript. We would like to thank the reviewers for their time and expertise in providing critical feedback in making this manuscript suitable for publication. We have revised our manuscript (Manuscript ID: ijms-2308307) according to the reviewers’ comments. In addition to these changes, we have also made substantial revisions to improve the style, flow, and clarity of this manuscript. We hope these changes improve the overall quality of this manuscript for publication. We have listed the reviewers’ comments and answered them in sequence.

Reviewer 1 Comments:

Comment: I read the paper with great interest. Despite you making a lot of effort in this review summarizing more than 80 up-to-date references, some modifications are required. The objective described in the title is not entirely addressed.

Response: We have shortened the title (Lines 2, 3). We feel that the choices most used for cannabis analysis are different than those used in the food industry. Further, we feel that the trend in food analysis to a more comprehensive analysis might lead to a different way of approaching cannabis analysis. We hope that by responding to the reviewers that we have addressed the objective described in the title.

Comment: The chemistry of the Cannabis plant is very complex and some important aspects we're not considered; for example, in table 2, you undervalued which part of the plant has been used for the extraction, which cultivar, etc.

Response: For Table 2 in Line 149, we discuss the analytical tools that have been used for the analysis of compounds found in cannabis products. We do agree that the plant part plays an important role in determining the best extraction method for subsequent analysis. All references in Table 2 have much greater detail on the extractions used, and the plant part studied.

Comment: Furthermore, there isn't any analysis or 'discussion' but only a lot of statements about what is in the literature with examples off-topic.

Response:  We hope that the improvements to the manuscript will help future readers more clearly see how examples are on-topic. The following sections have discussions and analyses of the topic.

Section 3 “Current system and potential quality issues” and Section 5 “Future directions” provides the necessary discussion on the quality issues and how to mitigate/ solve them using metabolomic/ big data approach.

In section 4 “Authentomics” we have included the explanation that cannabis could benefit from food analysis approaches.

***Paragraph on how this same level of analysis could be performed on cannabis

Lines 497-525

***Paragraph comparing targeted analysis on cannabis

Lines 553-562

***Paragraph on how cannabis could benefit from non-targeted analysis

Lines 595-602

***Paragraph on possible benefits of cannabis metabolome database and standard methods

Lines 644-655

Comment: I think you need to rethink the intention of your manuscript and reorganize it better.

Response: We have taken the editor's and reviewer's comments into careful consideration and have made significant revisions to our manuscript to better align with the intended purpose and improve the overall organization and clarity. In addition to addressing specific feedback provided, we have also focused on improving the style and flow of the manuscript. We appreciate the helpful feedback provided and hope that these changes meet the expectations of the editor and reviewers.

Reviewer 2 Report

Cannabis Quality- A Metabolomics and Big Data Approach to

Cannabis Authenticity (Authentomics)

Pramodkumar D. Jadhav, Youn Young Shim, Ock Jin Paek, Jung-Tae Jeon, Hyun-Je Park, Ilbum Park, Eui-Seong Park and Martin J. T. Reaney

 Reviewer’s comments:

Line 62: Table. 1: In chemical class column, for different type of cannabinoids, please write Cannabinoids type and remove all “type” from CBC, CBD, ……………..

In Table 1, Are these chemicals mainly in leaves, seeds, flowers? Which part of the plant?

Line 75-77: Not questioning this is in the literature, if you have to maintain the discussion of drug uses, please stick to FDA approved drugs such as Epidiolex (for the treatment of seizures associated with two rare and severe forms of epilepsy, Lennox-Gastaut syndrome and Dravet syndrome), Marinol and Syndros, to treat nausea and vomiting caused by cancer chemotherapy and anorexia associated with weight loss in AIDS patients. Also, Cesamet, which is indicated for nausea and vomiting induced by cancer chemotherapy.

Line 95: Revise “LC” to “High performance Liquid Chromatography (HPLC)” and in line 98, revise “High performance Liquid Chromatography (HPLC)”to “HPLC”

Line 266: revise “subjected to” to “injected onto”

Line 367: Please add “(on a dry weight basis)” after 0.3% THC.

Author Response

Thank you for your patience and recommendations for strengthening our manuscript. We would like to thank the reviewers for their time and expertise in providing critical feedback in making this manuscript suitable for publication. We have revised our manuscript (Manuscript ID: ijms-2308307) according to the editors’ and reviewers’ comments. In addition to these changes, we have also made substantial revisions to improve the style, flow, and clarity of this manuscript. We hope these changes improve the overall quality of this manuscript for publication. We have listed the reviewers’ comments and answered them in sequence.

Reviewer 2 Comments:

Comment: Line 62: Table. 1: In chemical class column, for different type of cannabinoids, please write Cannabinoids type and remove all “type” from CBC, CBD.

Response: Changed as recommended in Line 63.

Comment: In Table 1, Are these chemicals mainly in leaves, seeds, flowers? Which part of the plant?

Response: We have modified the table title to clarify (Line 63).

Comment: Line 75-77: Not questioning this is in the literature, if you have to maintain the discussion of drug uses, please stick to FDA approved drugs such as Epidiolex (for the treatment of seizures associated with two rare and severe forms of epilepsy, Lennox-Gastaut syndrome and Dravet syndrome), Marinol and Syndros, to treat nausea and vomiting caused by cancer chemotherapy and anorexia associated with weight loss in AIDS patients. Also, Cesamet, which is indicated for nausea and vomiting induced by cancer chemotherapy.

Response: The reviewer raised a valid concern, and we have addressed it in our article in Lines 76-82. While maintaining our original claim that some researchers believe that the association of cannabinoid structures with biological activity can lead to the development of cannabinoid-based drug formulations, we acknowledge the importance of citing only FDA-approved drugs such as Epidiolex, Marinol, Syndros, and Cesamet, which have been shown to effectively treat specific conditions.

Comment: Line 95: Revise “LC” to “High performance Liquid Chromatography (HPLC)” and in line 98, revise “High performance Liquid Chromatography (HPLC)”to “HPLC”

Response: Revised Lines 100, 101, and 104.

Comment: Line 266: revise “subjected to” to “injected onto”

Response: Revised Line 269.

Comment: Line 367: Please add “(on a dry weight basis)” after 0.3% THC.

Response: Revised Line 370.

Round 2

Reviewer 2 Report

All my comments have been clearly addressed in the revised version of the manuscript. No more comments. Thank you.